# Work-related stress, reason for consultation and diagnosis-specific sick leave: How do they add up?

**Anna-Maria Hultén**[1]*, **Kristina Holmgren**[1], **Pernilla Bjerkeli**[2]

**1** Unit of Occupational Therapy, Department of Health and Rehabilitation, Institute of Neuroscience and Physiology, Sahlgrenska Academy, University of Gothenburg, Gothenburg, Sweden, **2** Department of Public Health Research, University of Skövde, Skövde, Sweden

* anna-maria.hulten@gu.se

## Abstract

Work-related stress is common in Western society and disorders associated with stress are often managed in primary health care. This study was set to increase the understanding of the relationship between reason for consultation, work-related stress and diagnosis-specific sick leave for primary health care patients. The longitudinal observational study included 232 employed non-sick listed patients at seven primary health care centres in Sweden. Of these patients, 102 reported high work-related stress, as measured with the Work Stress Questionnaire, and 84 were on registered sick leave within one year after inclusion. The study showed that, compared to those who did not report high work-related stress, highly stressed patients more often sought care for mental symptoms (60/102 versus 24/130), sleep disturbance (37/102 versus 22/130) and fatigue (41/102 versus 34/130). The risk for sick leave with a mental diagnosis within a year after base-line was higher among patients reporting high work-related stress than among those who did not (RR 2.97, 95% CI 1.59;5.55). No such association was however found for the risk of sick leave with a musculo-skeletal diagnosis (RR 0.55, 95% CI 0.22;1.37). Seeking care for mental symptoms, sleep disturbance and fatigue were associated with having a future mental sick leave diagnosis (p-values < 0.001), while seeking care for musculoskeletal symptoms was associated with having a future musculoskeletal sick leave diagnosis (p-value 0.009). In summary, compared to those who did not report high work-related stress, patients with high work-related stress more often sought care for mental symptoms, sleep disturbance and fatigue which lead to a mental sick leave diagnosis. Reporting high work-related stress was, however, not linked to having sought care for musculoskeletal symptoms nor future sick leave due to a musculoskeletal diagnosis. Hence, both patients and general practitioners seem to characterize work-related stress as a mental complaint.

**Data Availability Statement:** To protect privacy and confidentiality, the raw data and the datasets generated in this study are not publicly available due to restrictions stated in the ethical approval (reference number 125-5) issued by the Regional

Ethical Review Board in Gothenburg, Sweden (later merged into the Swedish Ethical Review Authority, https://etikprovningsmyndigheten.se/). According to the approval, the data are to be managed carefully, and the results presented so that individuals cannot be identified. In addition, "All results from the project must be reported at group level with de-identified material" (ethical approval 125-5, section 6.3). The raw data and datasets generated cannot be made publicly available, because of the risk that individuals are identified if study data are combined. Sensitive data on work-related stress, days of sick leave and sick leave diagnosis could then be traced to the persons identified. Interested researchers may send data access requests to the corresponding author (AMH), anna-maria.hulten@gu.se, or the TIDAS research group at the Institute of neuroscience and physiology, tidas@neuro.gu.se. Please refer to the TIDAS project "Early identification of work-related stress", study "How do they add up?", and data set AddUp2023.

**Funding:** This study was funded by grant 2014-0936, received by KH, from the Swedish Research Council for Health, Working Life and Welfare (https://forte.se/). The funders had no role in study design, data collection and analysis, decision to publish, or preparation of the manuscript.

**Competing interests:** The authors have declared that no competing interests exist.

## Introduction

Stress, and especially work-related stress, is prevalent in the general population [1] and even more so among primary health care patients [2, 3]. Consequently, disorders that are associated with stress, such as depression, anxiety and stress-related disorders, are often managed within primary health care and the utilization of mental health care provided by primary health care has increased [4]. A common measure taken by general practitioners (GPs) to treat ill health due to stress and to handle reduced work capacity is to prescribe sick leave [5, 6]. Hence, the high prevalence of work-related stress among primary health care patients is reflected in the large number of individuals receiving sickness benefit with a mental disorder or a musculo-skeletal disease [7, 8]. In addition, the extent of sick leave is higher among employees on sick leave due to self-reported mental health problems compared to employees on sick leave due to other health reasons [9]. With this in mind, it is important to identify patients that could bene-fit from preventive measures in order not to risk long-term ill health and sick leave due to work related stress. A step in that direction is to understand the association between work-related stress, reason for consultation and sick leave diagnosis among primary health care patients.

In this study, work-related stress, reason for consultation and sick leave diagnosis are seen as parts of an illness-sickness-sick leave trajectory. Two underlying classification processes are important for the trajectory: the diagnosis process performed by the GP and the sick leave pro-cess handled by the sickness insurance officer and the GP [10]. The diagnosis process is ideally based on the GP's medical interpretation of the patient's anamnesis, risk factors, and symp-toms [10, 11]. In addition, the patient's reason for consultation has to be considered, since this can lead to different responses, diagnoses and management of diagnoses between GPs [12, 13]. Both the reason for consultation and the symptoms described by the patient are especially important for diagnoses when there are few or no clinical findings, as for mental disorders and musculoskeletal diseases such as musculoskeletal pain. Another important circumstance when diagnosing is that neither the patient nor the GP might be aware of or express that the symp-toms could be stress-related [3, 14]. However, in Sweden it is not the diagnosis as such, but the limitation in the patient's work capacity due to a disease or an injury that is the basis for the decision of whether a sick leave certificate is to be issued or not [15]. Even so, to support the decision of issuing sick leave, the National Board of Health and Welfare has authored recom-mendations on the assessment of work capacity for various diagnoses [15], thereby indirectly linking diagnosis to sick leave [16].

The International Classification of Diseases (ICD) is used globally in health care to classify and statistically describe sickness and health issues [17]. In Sweden, the ICD is widely used in outpatient and inpatient care, health insurance and occupational health care for information, communication and in registers as well as for business description and quality controls [18]. Work-related stress can be defined as "the response people may have when presented with work demands and pressures that are not matched to their knowledge and abilities and which challenge their ability to cope" [19]. Although stress is not classified as a disease or a disorder, being exposed to stressors or perceiving stress at work has been associated with adjustment disorder, major depressive disorder, generalized anxiety disorder [20, 21], gastrointestinal dis-orders [22], self-reported musculoskeletal pain [23], eczema [24] and coronary heart disease [25]. Therefore, several of the chapters included in the ICD are relevant when studying nega-tive health effects associated with work-related stress.

Individuals who work in Sweden are included in the Swedish social insurance system and thereby entitled to work-based benefits. When an individual becomes ill, the employer is responsible for paying sick pay to the employee during the first 14 days of sickness with one

qualifying day. If the work capacity continues to be impaired due to the sickness after 14 days, sickness cash benefits are handled by the Swedish Social Insurance Agency. From day eight of a sickness period, a medical certificate issued by a doctor is needed to certify that the work capacity is reduced. The certificate holds information on diagnosis, disability, activity limitation and work capacity as well as information on prognosis, treatment and measures to facilitate return to work. The information in the medical certificate forms the basis for the Swedish Social Insurance Agency's decision on sick leave. Approximately 625,000 Swedes (corresponding to 6% of the population) received sickness cash benefit on some occasion during 2016 [26], that is the year the follow up data for this this study was collected. Mental disorders were the most common type of diagnoses with the highest frequency among women (53%) and persons under fifty years of age. Musculoskeletal conditions were also common, especially among men (24%) and persons above fifty years of age [26].

The overall consensus is that the employee's cognitive, emotional and behavioural reaction to the social and organizational work context is important for staying healthy or becoming sick [27–33]. Herein, proximal job task characteristics, such as work pace and repetitive tasks, to labour market arrangements can affect the risk of perceiving work-related stress and ill health thereof [34–36]. Due to this complexity, different general social and organisational work characteristics such as efforts, demands, decision authority and organisational justice have been used as summative measures of the psychosocial risk factors at work [30], often in relation to a stress theory [35]. Even so, findings show that other factors also have to be accounted for. For instance, having a high degree of personal commitment to work whilst also perceiving the extrinsic aspects of work as stressful can affect future mental health and sick leave negatively [29, 37]. In addition, the interference between an individuals' working life and private life is a risk factor for later sick leave [38]. Further, job control reduces sick leave directly, but also indirectly through motivation [39]. In this study, the social and organizational conditions at work as well as personal commitment and interference between work and leisure time were therefore seen as important for when and to what extent individuals perceive stress to a level exceeding their room for manoeuvre and capacity to cope and thereby also for the association between work-related stress and sick leave.

In view of the above, the overall aim of this study was to understand the relationship between reason for consultation, work-related stress and diagnosis-specific sick leave for primary health care patients seeking care for mental and physical health complaints. By focusing on mental disorders and musculoskeletal diseases it was possible to include the most important diagnosis groups for stress-related ill health, whilst also including the majority of the certificates issued [8, 40]. The study was designed to answer the following research questions concerning primary health care patients of working age seeking care for mental and/or physical health complaints (see Fig 1 for summary):

How is the patient's reason for consultation related to the level of self-reported work-related stress?

Is the level of self-reported work-related stress at baseline associated with being sick listed with a future mental or a musculoskeletal sick leave diagnosis in the following 12 months?

Is the reason for consultation at baseline associated with being sick listed with a future mental or a musculoskeletal sick leave diagnosis in the following 12 months?

Are there differences in the association between reason for consultation and future sick leave diagnosis for patients perceiving high versus low work-related stress?

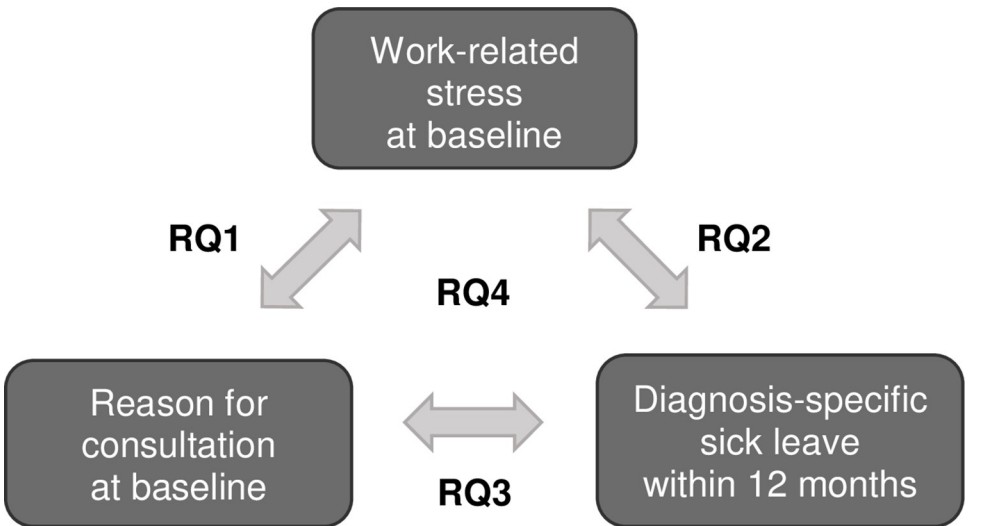

**Fig 1. Summary of the four associations and the related research questions (RQ 1–4).** Number 1–3 describe the bivariate associations, while number 4 includes all three variables (work-related stress, reason for consultation and diagnosis-specific sick leave).

## Materials and methods

The associations between work-related stress, as measured with the Work Stress Questionnaire (WSQ) [41], reason for consultation and diagnosis-specific sick leave were investigated for primary health care patients in the Västra Götaland region in Sweden in this prospective longitudinal observational study with initial cross-sectional analysis.

### Setting and study population

The study was conducted at seven primary health care centres located in both urban and rural areas of the region (Table 1). Included in the study were primary health care patients participating in a randomized controlled trial (RCT) (ClinicalTrials.gov, identifier: NCT02480855, ethical approval 125–15 from the Regional Ethical Review Board in Gothenburg, Sweden). The RCT aimed to evaluate the effectiveness of a brief intervention targeting primary health care

**Table 1. Primary health care centre characteristics and inclusion of participants.**

| Primary health care centre | | | Primary health care patients | | | | |
|---|---|---|---|---|---|---|---|
| No. | Provision | Location[1] | Eligible | Declined | Ex-cluded[2] | Included in RCT | Included in study[3] |
| 1 | Public | Accessible small town | 103 | 10 | 6 | 87 | 79 |
| 2 | Public | Accessible rural area | 9 | 0 | 1 | 8 | 8 |
| 3 | Private | Large urban area | 20 | 2 | 0 | 18 | 18 |
| 4 | Public | Other urban area | 57 | 3 | 3 | 51 | 42 |
| 5 | Public | Accessible small town | 30 | 0 | 1 | 29 | 23 |
| 6 | Public | Other urban area | 48 | 3 | 1 | 44 | 34 |
| 7 | Private | Large urban area | 36 | 2 | 0 | 34 | 28 |
| | | Total | 303 | 20 | 12 | 271 | 232 |

[1] Classified according to the Urban Rural classification [46].

[2] Patients who gave their consent to participate, but was not included in the RCT due to logistic reasons.

[3] Patients who completed the Work Stress Questionnaire.

patients seeking care for mental and physical health complaints [42]. Based on three outcome evaluations, the brief intervention showed no effect on the number of self-reported or the number of registered sick leave days [43–45]. Hence, the data collected in the RCT were considered useful in this longitudinal observational study to analyse the relationship between factors observed at baseline and future sick leave without regard to the intervention given.

Patients included had to be employed, non-sick-listed, 18–64 years of age and seeking care for mental and/or physical health complaints. These complaints included depression, anxiety, musculoskeletal disorders, gastrointestinal and cardiovascular symptoms as well as other symptoms that could be stress-related. In addition, the questionnaire assessing work-related stress had to be completed. Therefore, 232 patients participated in the study in hand (Table 1). Patients seeking care for other reasons such as diabetes, infections, fractures, lumps and spots, allergy, psychoses and medical check-ups were excluded. Pregnant women, patients currently on sick leave and those who had been on sick leave for 7 days or more during the last month as well as those with a disability pension were also excluded. Baseline data were collected from May 2015 to January 2016. The patients were given oral and written information about the trial. In addition, written informed consent for participating in the trial was provided.

## Data sources

When seeking care at the primary health care centre, the patients filled out a questionnaire to provide baseline data on self-assessed work-related stress, reason for consultation and background characteristics. Sick leave data were retrieved from the national database MiDAS (Micro Data for Analysis of Social Insurance), which is held by the Swedish Social Insurance Agency. The database includes information on all sick leave spells exceeding 14 days issued in Sweden. In this study, data on the patients' sick leave diagnoses were retrieved for each spell of sick leave exceeding 14 days within 12 months of inclusion.

## Variables

**Work-related stress.** The WSQ [41] was used to assess the individual's own perception of having stressors and the levels of work-related stress thereof. The questionnaire includes 21 items with 14 appended items covering personal characteristics as well as the working and private life, to capture the work situation (S1 Appendix). The items are grouped in four dimensions:

- Influence at work, captures the employee's possibilities to influence the performance of work tasks as well as the work situation in general (item 1–4). The four items are answered on four-point ordinal scale scales with response alternatives 'Yes, always', 'Yes, rather often', 'No, seldom' and 'No, never';

- Perceived stress due to indistinct organization and conflicts, contains questions concerning work load, the clarity of goals, roles and assignments as well as involvement and handling of conflicts. If experiencing any indistinct organization or conflicts (item 5a–11a), the perceived stress thereof (item 5b–11b) is answered on four-point ordinal scales with response alternatives 'Not at all stressful', 'Less stressful', 'Stressful', and 'Very stressful';

- Perceived stress due to individual demands and commitment contains questions concerning for example putting on too much pressure, working beyond working hours and setting limits. If experiencing any individual demands and commitment (item 12a–18a), the perceived stress thereof (item 12b–18b) is answered on four-point ordinal scales with response alternatives 'Not at all stressful', 'Less stressful', 'Stressful', and 'Very stressful';

- Work to leisure time interference, captures how time spent with nearest, friends and recreational activities is affected by work (item 19–21). The three items are answered on four-point ordinal scale with response alternatives 'Yes, always', 'Yes, rather often', 'No, seldom' and 'No, never'.

The face validity and the test-retest reliability of the questionnaire have been tested for women and men separately and found satisfying [41, 47].

Based on the patient's score for each item, a median value for each of the four dimensions was calculated as described in the WSQ instructions (S1 Appendix). The median values were then dichotomized into high and low exposure with the value 2.0 as a cut-off. Earlier research [3, 48] have shown that having work-related stressors or perceiving stress thereof within multiple areas could increase the odds of future sick leave. The number of dimensions indicating high work-related stress was therefore used as a summative value of the exposure to work-related stress. The work-related stressors and stress were operationalized with five measures:

- Influence at work (dimension 1, item 1–4) was summarized in two categories based on the median value: low, > 2 (No, never or No, seldom) and high, ≤ 2 (Yes, often or Yes, always);

- Perceived stress due to indistinct organization and conflicts (dimension 2, item 5b–11b) was summarized in two categories based on the median value: high, > 2 (Stressful or Very stressful) and low, ≤ 2 (not experienced, Not stressful or Less stressful);

- Perceived stress due to individual demands and commitment (dimension 3, item 12b–18b) was summarized in two categories based on the median value: high, > 2 (Stressful or Very stressful) and low, ≤ 2 (not experienced, Not stressful or Less stressful);

- Work to leisure time interference (dimension 4, item 19–21) was summarized in two categories based on the median value: high, > 2 (Yes, often or Yes, always) or low, ≤ 2 (no, never or no, seldom);

- The number of dimensions indicating high stress was summarized in the two categories: having 0–1 dimension with median value > 2 and having 2–4 dimensions with median value > 2.

**Reason for consultation.** The reason for consultation was captured with the question, 'What complaints are you seeking care for today?'. Fifteen response alternatives were given, including a free text response, and multiple answers were possible. Seven dichotomized measures were formulated based on these responses:

- Seeking care for mental symptoms including stress, anxiety, depression and other mental symptoms (yes or no).

- Seeking care for musculoskeletal symptoms including neck, shoulder and other musculoskeletal symptoms (yes or no).

- Seeking care for sleep disturbance (yes or no).

- Seeking care for fatigue (yes or no).

- Seeking care for gastrointestinal symptoms (yes or no).

- Seeking care for cardiovascular symptoms (yes or no).

- Seeking care for other symptoms (yes or no).

**Diagnosis-specific sick leave.** The main sick leave diagnosis included in the sickness certificate was coded according to the Swedish version of the International Classification of Diseases, 10th Revision (ICD-10) [18]. The diagnosis-specific sick leave was characterized with the following measure:

- Sick leave with mental diagnosis, defined as having had at least one spell of registered sick leave (>14 days long) with a mental diagnosis (ICD-10, chapter F) within 12 months after baseline (yes or no).

- Sick leave with musculoskeletal diagnosis, defined as having had at least one spell of registered sick leave (>14 days long) with a musculoskeletal diagnosis (ICD-10; chapter M) within 12 months after baseline (yes or no).

Cases with rehabilitation benefits and preventive sickness benefits were excluded.

**Background variables.** The background variables were selected to describe the individual's demographics, socioeconomic position, sick leave history and terms of employment.

- Sex was measured as a nominal variable with the two categories women and men.

- Age was measured as a continuous variable and then transformed into an ordinal variable with the three age groups 18–30, 31–50 and 51–64 years of age.

- Educational level was measured as an ordinal variable with the alternatives elementary school not completed, elementary school, high school 2 years, high school 3–4 years, university less than 3 years and university 3 years or more. The categories were then summarized in the two categories university and others.

- Occupational class was measured as a nominal variable with three categories; high-level non-manual, medium/low non-manual and skilled/unskilled manual. The categorization of the respondent's occupation was performed according to Statistics Sweden's socioeconomic classification of persons in the labour force [49].

- Marital status was measured as a nominal variable with three categories and then summarized in two categories: not single (married/cohabitant or living apart) and single.

- Previous registered sick leave was chosen to represent the sickness status before the study and it was quantified with the measure: Having had a spell of sick leave exceeding 14 days in the year before baseline (yes or no).

- Employer was measured as a nominal variable with six categories and then summarized in the two categories public employer and private employer.

**Statistical analyses.** Descriptive statistics were used to summarize the socio-demographics, the sickness history and the terms of employment for the total study population as well as for patients with high work-related stress. The Pearson chi-squared test was then used to evaluate the association between reason for consultation and work-related stress (research question 1) and the association between future mental or a musculoskeletal sick leave diagnosis and self-assessed work-related stress (research question 2). In addition, the relative risks (RR) for future mental or musculoskeletal sick leave diagnoses were calculated for patients perceiving high compared to low work-related stress (research question 2). The association between the reason for consultation and sick leave diagnosis was tested using Pearson chi-squared test for both the study population in total (research question 3) as well as for a stratified sample

including patients perceiving high stress (research question 4). Patients perceiving high and low work-related stress were then analysed separately.

Having stressors or perceiving work-related stress in 2–4 WSQ dimensions was used as an indicator of high work-related stress. In general, when 20% or more of the cells in the contingency table had an expected cell frequency under the null hypothesis of less than 5, the Fisher's exact test was used instead of the Pearson chi-squared test to calculate the association between two variables [34]. Irrespective of analysis, the level of statistical significance was set to p-value ≤ 0.05. The statistical analyses were performed using IBM SPSS Statistics version 25.0.

## Results

### Background characteristics and work-related stress

As seen in Table 2, the study included 232 primary health care patients of which 153 (66%) were women and 79 (34%) were men. Fifty percent of this working age population were between 31–50 years of age, with an average age of 44 years. In addition, having a university degree or a high school degree was equally common; 44% versus 46%. Moreover, 102 (44%) of the patients worked in the public sector, while 130 (56%) of the patients worked in the private sector. Statistically significant differences for patients reporting high versus low work-related stress were seen in relation to sex (p-value 0.007), with a higher proportion of women reporting high stress, but also in relation to educational level (p-value 0.025), where work-related stress was more frequent among patients with a university degree than among those who did not have a university degree.

**Table 2. Background characteristics for the total study population and the subgroup perceiving high work-related stress.**

| Variable | | Total | | High work-related stress[1] | | |
|---|---|---|---|---|---|---|
| | | n | % | n | % | p-value[2] |
| Total | | 232 | | 102 | 44 | |
| Sex | Women | 153 | 66 | 77 | 50 | **0.007** |
| | Men | 79 | 34 | 25 | 32 | |
| Age | 18–30 | 41 | 18 | 17 | 41 | 0.639 |
| | 31–50 | 117 | 50 | 55 | 47 | |
| | 51–64 | 74 | 32 | 30 | 41 | |
| Educational level[3] | University | 103 | 44 | 55 | 53 | **0.025** |
| | High school | 106 | 46 | 41 | 39 | |
| | Elementary school | 22 | 10 | 6 | 27 | |
| Occupational class[3] | High-level non-manual | 42 | 18 | 20 | 48 | 0.503 |
| | Medium/low non-manual | 100 | 43 | 47 | 47 | |
| | Skilled/unskilled manual | 89 | 38 | 35 | 39 | |
| Marital status[4] | Not single | 185 | 80 | 78 | 42 | 0.176 |
| | Single | 45 | 19 | 24 | 53 | |
| Registered sick leave prior year | Yes | 28 | 11 | 12 | 43 | 0.900 |
| | No | 204 | 89 | 90 | 44 | |
| Employer | Public | 102 | 44 | 50 | 49 | 0.057 |
| | Private | 130 | 56 | 52 | 40 | |

[1] Perceiving stressors or stress within at least two of the four dimensions included in the Work Stress Questionnaire.

[2] Pearson chi-squared test for patients having high compared to low work-related stress.

[3] One missing value

[4] Two missing values

**Table 3. Relationship between symptoms given as reason for consultation and work-related stress (N = 232).**

| Symptoms[1] | | Total | | High work-related stress[2] | | |
|---|---|---|---|---|---|---|
| | | n[1] | % | Yes (n, %) | No (n, %) | p-value[3] |
| Total | | 232 | | 102 (44) | 130 (56) | |
| Mental symptoms[4] | Yes | 84 | 36 | 60 (71) | 24 (29) | < **0.001** |
| | No | 148 | 64 | 42 (28) | 106 (72) | |
| Musculoskeletal symptoms[5] | Yes | 88 | 38 | 34 (39) | 54 (61) | 0.201 |
| | No | 144 | 62 | 68 (47) | 76 (53) | |
| Sleep disturbance | Yes | 59 | 25 | 37 (63) | 22 (37) | **0.001** |
| | No | 173 | 75 | 65 (38) | 108 (62) | |
| Fatigue | Yes | 75 | 32 | 41 (55) | 34 (45) | **0.023** |
| | No | 157 | 68 | 61 (39) | 96 (61) | |
| Gastrointestinal symptoms | Yes | 45 | 19 | 20 (44) | 25 (56) | 0.943 |
| | No | 187 | 81 | 82 (44) | 105 (56) | |
| Cardiovascular symptoms | Yes | 25 | 11 | 11 (44) | 14 (56) | 0.997 |
| | No | 207 | 89 | 91 (44) | 116 (56) | |
| Other symptoms | Yes | 46 | 20 | 23 (50) | 23 (50) | 0.357 |
| | No | 186 | 80 | 79 (43) | 107 (57) | |

[1] Selecting multiple symptoms was optional

[2] Perceiving stressors or stress within at least two of the four dimensions included in the Work Stress Questionnaire.

[3] Pearson chi-squared test for patients having high compared to low work-related stress.

[4] Stress, anxiety, depression and other mental symptoms

[5] Neck, shoulder and other musculoskeletal symptoms

## Reason for consultation and work-related stress

Musculoskeletal symptoms were reported as a reason for consultation by 38% (88/232) patients in the population and almost as many, 36% (84/232) patients, reported mental symptoms as a reason for consultation. The reason for consultation was associated with self-assessed work-related stress. As seen in Table 3, there is a statically significant difference in in the proportion of patients reporting high work-related stress for patients who sought care for mental symptoms such as stress, anxiety and depression compared to those who sought care for other reasons. Such differences were also seen among patients who sought care for sleep disturbance and fatigue.

## Work-related stress and diagnosis-specific sick leave

In the study population, 36% (84/232) were on registered sick leave during the year following baseline. Sixty-nine patients had sick leave linked to one diagnosis, while 11 patients had sick leave linked to 2–3 different diagnoses. In addition, four patients were on sick leave with an unspecified diagnosis. For 40 (48%) of the 84 patients, mental and behavioural disorders such as depression, anxiety and adjustment disorders were reported as a sick leave diagnosis (Table 4). Diseases of the musculoskeletal system and connective tissue such as arthrosis, dorsopathies and soft tissue disorders were reported as a sick leave diagnosis for 20 (24%) of the 84 patients. Twenty-seven patients had a sick leave diagnoses within fourteen other ICD-10 chapters.

In Table 5, work-related stress is compared to future mental or musculoskeletal sick leave diagnosis. Noteworthy is that 28 (27%) of the 102 patients perceiving stressors and stress within multiple dimensions, had a mental sick leave diagnosis within one year after baseline,

**Table 4. Frequency of diagnosis-specific sick leave classified according to the International Classification of Diseases, Tenth Revision (ICD-10) within one year of baseline (N = 84).**

| Sick leave diagnosis (ICD-10 chapter) | Category | Number of patients[1] |
|---|---|---|
| Mental and behavioural disorder (F00-F99) | F19, F32, F33, F41, F43 | 40 |
| Diseases of the musculoskeletal system and connective tissue (M00-M99) | M18, M25, M50, M51, M54, M75, M76, M77, M79 | 20 |
| Other ICD—chapters (A, B, D, E, G, J, K, L, N, O, R, S, T and Z) | | 27 |
| Diagnosis missing or unknown | | 4 |

[1] Patients could have sick leave diagnoses within multiple chapters during 12 months after baseline.

while 12 (9%) of the 130 patients perceiving stressors and stress within zero to one dimension, had a mental sick leave diagnosis within one year after baseline. However, 74 (73%) of the 102 patients perceiving stressors and stress within multiple dimensions did not have any future

**Table 5. Relationship between work-related stress as measured with the Work Stress Questionnaire and future mental sick leave diagnosis or musculoskeletal sick leave diagnosis (N = 232).**

| Work-related stress | | Mental sick leave diagnosis | | | |
|---|---|---|---|---|---|
| | | Yes | No | p-value[1] | RR (95% CI)[2] |
| Total | | 40 | 192 | | |
| Influence at work | Low | 24 | 70 | **0.006** | **2.20 (1.24;3.92)** |
| | High | 16 | 122 | | 1.00 |
| Stress due to organisation and conflicts | High | 12 | 37 | 0.130 | 1.60 (0.88;2,91) |
| | Low | 28 | 155 | | 1.00 |
| Stress due to demands and commitment | High | 28 | 77 | **0.001** | **2.82 (1.51;5.27)** |
| | Low | 12 | 115 | | 1.00 |
| Work interference with leisure time | High | 23 | 69 | **0.011** | **2.06 (1.16;3.64)** |
| | Low | 17 | 123 | | 1.00 |
| Number of dimensions with high stress | 2–4 dim | 28 | 74 | **< 0.001** | **2.97 (1.59;5.55)** |
| | 0–1 dim | 12 | 118 | | 1.00 |
| **Work-related stress** | | **Musculoskeletal sick leave diagnosis** | | | |
| | | Yes | No | p-value[1] | RR (95% CI)[2] |
| Total | | 20 | 212 | | |
| Influence at work | Low | 7 | 87 | 0.599 | 0.79 (0.33;1.91) |
| | High | 13 | 125 | | 1.00 |
| Stress due to indistinct organisation and conflicts | High | 5 | 44 | 0.774[3] | 1.24 (0.48;3.26) |
| | Low | 15 | 168 | | 1.00 |
| Stress due to individual demands and commitment | High | 8 | 97 | 0.621 | 0.81 (0.34;1.90) |
| | Low | 12 | 115 | | 1.00 |
| Work interference with leisure time | High | 9 | 83 | 0.609 | 1.24 (0.54;2.89) |
| | Low | 11 | 129 | | 1.00 |
| Number of dimensions with high stress | 2–4 dim | 6 | 96 | 0.188 | 0.55 (0.22;1.37) |
| | 0–1 dim | 14 | 116 | | 1.00 |

[1] Pearson chi-squared test for patients having high compared to low work-related stress

[2] Relative risk and 95% confidence interval

[3] Fisher's exact test was used instead of Pearson chi-squared test to calculate the p-value, since the expected cell frequency was less than 5 in more than 20% of the cells in the contingency table.

mental sick leave diagnosis. Corresponding findings were seen when studying the dimensions influence at work, high stress due to individual demands and commitment as well as interference between work and leisure. The relative risk for having a future mental sick leave diagnosis when perceiving high work-related stress is also presented in Table 5. Patients perceiving low influence at work, high stress due to demands and commitment or high interference between work and leisure had an increased risk of having a future mental sick leave diagnosis (RR 2.20, 2.82 and 2.06 respectively). However, the largest risk increase was seen among patients perceiving stress within multiple stress dimensions (RR 2.97). Moreover, no statistically significant association or increased risk was seen when comparing work-related stress and future musculoskeletal sick leave diagnosis.

### Reason for consultation and diagnosis-specific sick leave

The results from the analysis of the relationship between the symptoms given as the reason for consultation and future diagnosis-specific sick leave are shown in Table 6. Mental symptoms as well as sleep disturbance and fatigue, have a statistically significant association with having a future mental sick leave diagnosis (p-values < 0.001). In addition, there is a significant association between seeking care for musculoskeletal symptoms and having a future musculoskeletal sick leave diagnosis (p-value 0.009). In summary, the patients' diagnoses and their reason for consultation were matching. However, 11 (27%) of the 40 patients who had a future mental sick leave diagnosis did not express any mental symptoms at baseline. In addition, 55 (65%) of the 84 patients who sought care for mental symptoms did not have any mental sick leave diagnosis within one year after baseline.

**Table 6. Relationship between symptoms given as reason for consultation and having a mental or musculoskeletal sick leave diagnosis (N = 232).**

| Symptoms[1] | | Total | Mental sick leave diagnosis N = 40 | | | Musculoskeletal sick leave diagnosis N = 20 | | |
|---|---|---|---|---|---|---|---|---|
| | | n | Yes | No | p-value[2] | Yes | No | p-value[2] |
| Total | | 232 | 40 | 192 | | 20 | 212 | |
| Mental symptoms[3] | Yes | 84 | 29 | 55 | < **0.001** | 3 | 81 | 0.039 |
| | No | 148 | 11 | 137 | | 17 | 131 | |
| Musculoskeletal symptoms[4] | Yes | 88 | 14 | 74 | 0.675 | 13 | 75 | **0.009** |
| | No | 144 | 26 | 118 | | 7 | 137 | |
| Sleep disturbance | Yes | 59 | 20 | 39 | < **0.001** | 5 | 54 | 0.963 |
| | No | 173 | 20 | 153 | | 15 | 158 | |
| Fatigue | Yes | 75 | 28 | 47 | < **0.001** | 5 | 70 | 0.464 |
| | No | 157 | 12 | 145 | | 15 | 142 | |
| Gastrointestinal symptoms | Yes | 45 | 7 | 38 | 0.739 | 4 | 41 | 1.000[5] |
| | No | 187 | 33 | 154 | | 16 | 171 | |
| Cardiovascular symptoms | Yes | 25 | 6 | 19 | 0.344 | 1 | 24 | 0.705[5] |
| | No | 207 | 34 | 173 | | 19 | 188 | |
| Other symptoms | Yes | 46 | 5 | 41 | 0.201 | 4 | 42 | 1.000[5] |
| | No | 186 | 35 | 151 | | 16 | 170 | |

[1] Selecting multiple symptoms was possible

[2] Pearson chi-squared test for patients having high compared to low work-related stress

[3] Stress, anxiety, depression and other mental symptoms

[4] Neck, shoulder and other musculoskeletal symptoms

[5] Fisher's exact test was used instead of Pearson chi-squared test to calculate the p-value, since the expected cell frequency was less than 5 in more than 20% of the cells in the contingency table.

### Reason for consultation, diagnosis-specific sick leave and work-related stress

A stratified analysis of the relationship between the reason for consultation and future sick leave diagnosis was performed for patients perceiving high work-related stress, that is reporting high stress within multiple WSQ dimensions. As seen in Table 7, the relationship between mental symptoms and future mental sick leave diagnosis presented in Table 6 still holds (p-value 0.001), even if the frequencies have changed. In this stratified sample, 14% (4/28) of the patients had a future mental sick leave diagnosis even if they did not express any mental symptoms at baseline compared to 27% (11/40) in the total study population. For patients perceiving high work-related stress, no association was found between the reason for consultation and having a future musculoskeletal sick leave diagnosis, which is in contrast to the total study sample.

Fig 2 shows that, among the 60 highly stressed patients who sought care for mental symptoms, 24 (40%) had a future mental sick leave diagnosis, while the corresponding value for patients perceiving low stress was 5 out of 24 (21%).

Fig 3 shows that, among the 34 highly stressed patients who sought care for musculoskeletal symptoms, 2 (6%) had a future musculoskeletal sick leave diagnosis, while the corresponding value for patients perceiving low stress was 11 out of 54 (20%). That is, both the degree of perceived work-related stress and the reason for consultation were important for future sick leave diagnosis.

**Table 7. Relationship between symptoms given as reason for consultation compared to having a mental or musculoskeletal sick leave diagnosis for patients perceiving high work-related stress (N = 102).**

| Symptoms[1] | | Total | Mental diagnosis | | | Musculoskeletal diagnosis | | |
|---|---|---|---|---|---|---|---|---|
| | | n[2] | Yes | No | p-value[3] | Yes | No | p-value[4] |
| Total | | 102 | 28 | 74 | | 6 | 96 | |
| Mental symptoms[5] | Yes | 60 | 24 | 36 | **0.001** | 3 | 57 | 0.688 |
| | No | 42 | 4 | 38 | | 3 | 39 | |
| Musculoskeletal symptoms[6] | Yes | 34 | 9 | 25 | 0.875 | 2 | 32 | 1.000 |
| | No | 68 | 19 | 49 | | 4 | 64 | |
| Sleep disturbance | Yes | 37 | 16 | 21 | **0.007** | 5 | 32 | 0.023 |
| | No | 65 | 12 | 53 | | 1 | 64 | |
| Fatigue | Yes | 41 | 21 | 20 | **< 0.001** | 4 | 37 | 0.216 |
| | No | 61 | 7 | 54 | | 2 | 59 | |
| Gastrointestinal symptoms | Yes | 20 | 5 | 15 | 0.784 | 1 | 19 | 1.000 |
| | No | 82 | 23 | 59 | | 5 | 77 | |
| Cardiovascular symptoms | Yes | 11 | 3 | 8 | 1.000[4] | 1 | 10 | 0.505 |
| | No | 91 | 25 | 66 | | 5 | 86 | |
| Other symptoms | Yes | 23 | 4 | 19 | 0.219 | 2 | 21 | 0.615 |
| | No | 79 | 24 | 55 | | 4 | 75 | |

[1] Selecting multiple symptoms was optional

[2] Number of patients perceiving stressors or stress within at least two of the four dimensions included in the Work Stress Questionnaire.

[3] Pearson chi-squared test for patients having high compared to low work-related stress

[4] Fisher's exact test was used instead of Pearson chi-squared test to calculate the p-value, since the expected cell frequency was less than 5 in more than 20% of the cells in the contingency table.

[5] Stress, anxiety, depression and other mental symptoms

[6] Neck, shoulder and other musculoskeletal symptoms

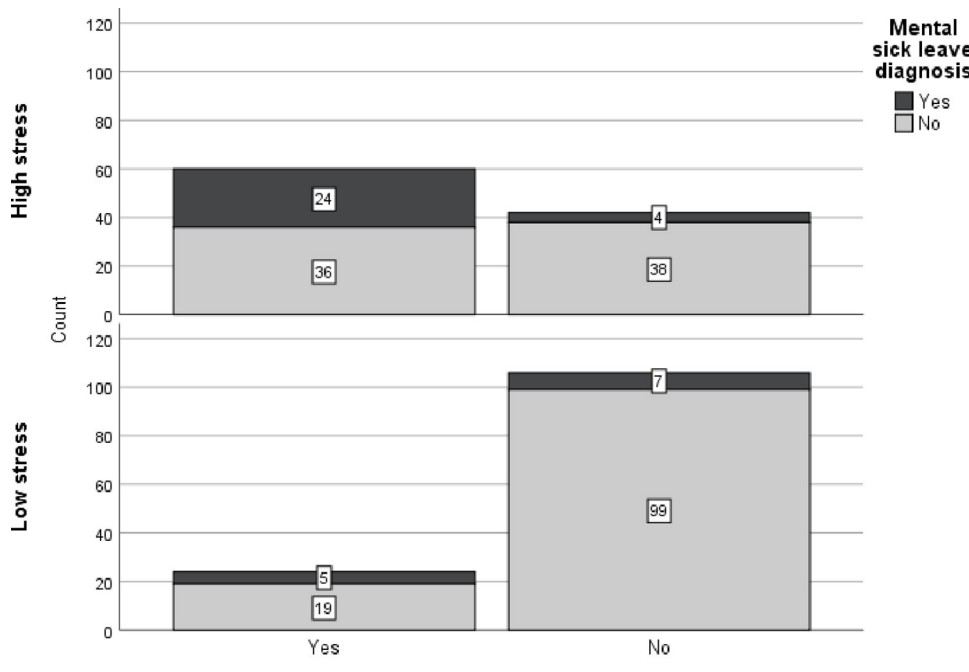

**Fig 2. Frequency of mental diagnoses for patients seeking care for mental or others reasons.** Separate graphs for patients perceiving high versus low work-related stress.

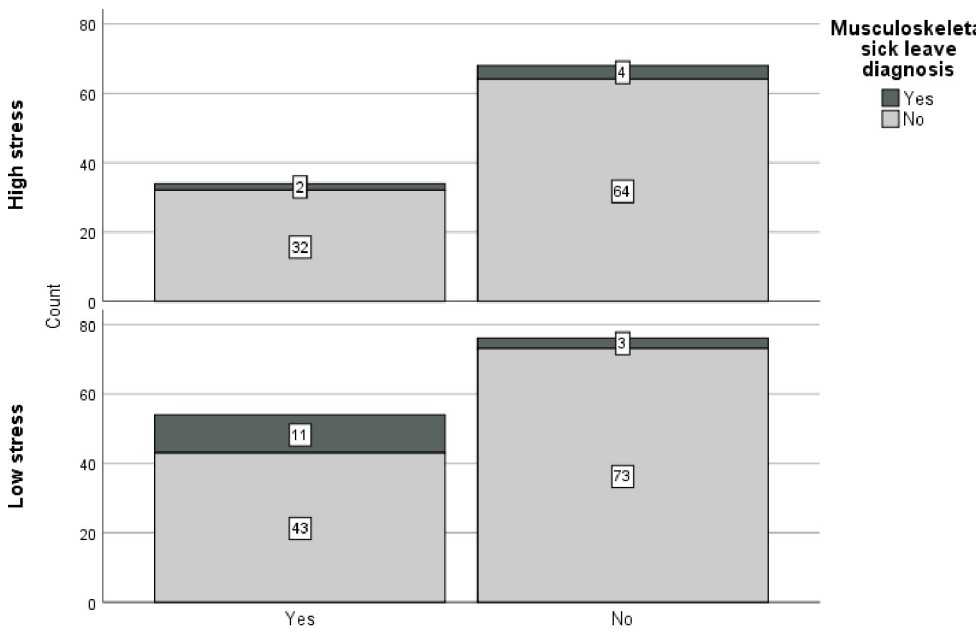

**Fig 3. Frequency of musculoskeletal diagnoses for patients seeking care for musculoskeletal or other reasons with separate graphs for patients perceiving high versus low work-related stress.**

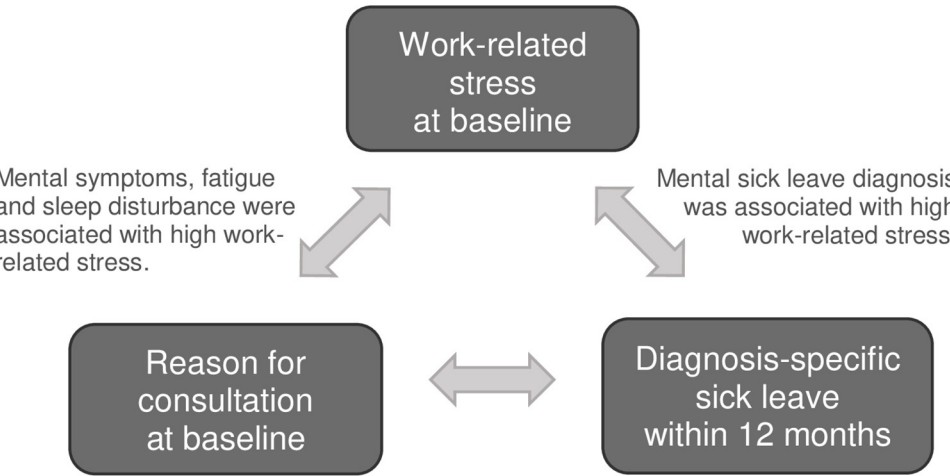

**Fig 4. Summary of the study findings for the three variables work-related stress, reason for consultation and diagnosis-specific sick leave.** The text boxes describe the main results for the associations marked with arrows.

## Discussion

### Principal findings

This longitudinal observational study showed that primary health care patients with high work-related stress more often sought care for mental symptoms, sleep disturbance and fatigue compared to those who did not report high work-related stress (Fig 4). The most common reason for consultation among patients reporting high work-related stress was mental symptoms. The risk of having a future mental sick leave diagnosis within a year after baseline was higher among patients reporting high work-related stress. The concordance between patients' reason for consultation and their future sick leave diagnosis was relatively good. For example, among patients who sought care for a mental symptom it was more common to have a future mental sick leave diagnosis than not to, and correspondingly, among patients who had a mental sick leave diagnosis, a majority had sought care for mental symptoms within the previous year. This association was seen in the study population as a whole as well as in the stratified sample of patients perceiving high work-related stress. In addition, patients seeking care for musculoskeletal symptoms more often had a future musculoskeletal sick leave diagnosis than those who sought care for other reasons, but this association was only seen in the study population as a whole.

### Interpretation of findings

A prior study performed by Hultén et al. [48] showed that work-related stress was common among primary health care patients and that one third of the patients were on registered sick leave within a year after inclusion. In addition, work-related stress increased the odds of future registered sick leave, thereby confirming prior research performed in a primary health care setting [3]. The study in hand examined the association between work-related stress and sick leave among primary health care patients in more detail by focusing on diagnosis-specific sick leave and including the reason for consultation in the analysis.

The reason for consultation was in this study found to be associated with work-related stress. Among the 102 patients with high work-related stress, 60% sought care for mental symptoms, while more than one third reported musculoskeletal symptoms, fatigue and/ or sleep disturbance as a reason for consultation, that is symptoms known to be associated with stress [50]. In a similar study, Wiegner et al. [2] found that among highly stressed Swedish primary health care patients, one third reported symptoms that could be depression-related and two thirds reported possibly anxiety-related symptoms. In another Swedish study [14], patients with exhaustion disorder were found to consult their GP for different complaints that could be stress-related, most frequently infection, anxiety/depression and stress, in the years preceding their diagnosis. Hence, the symptoms presented as a reason for consultation can contribute to an understanding of why patients with high work-related stress chose to contact primary health care.

Perceiving low influence at work, high stress due to demands and commitment or high interference between work and leisure more than doubled the risk of having a future mental sick leave diagnosis in this study. The results thus support prior studies on the association between work-related stress and sick leave [3, 29, 48]. In addition, Duchaine et al. [51] recently performed a systematic review and meta-analysis of 13 studies showing that working adults perceiving high demands and low control at work had an almost 50% increased risk of sick leave due to a mental disorder. The results are also in line with studies examining the relationship between psychosocial risk factors at work and mental diagnoses. In a systematic review [30], van der Molen et al. concluded that social and organisational factors at work, and in particular effort-reward imbalance, low organisational justice and high job demands, increased the risk of stress-related mental disorders among the working population. However, the study in hand showed no association between work-related stress and musculoskeletal sick leave diagnoses, which in part contradicts prior study findings on the relationship between psychosocial risk factors at work and musculoskeletal diagnoses [27, 32].

The reason for consultation was in this study associated with future sick leave diagnosis, that is seeking care for mental symptoms, fatigue or sleep disturbance was positively associated with a mental sick leave diagnosis while seeking care for musculoskeletal symptoms prompted a musculoskeletal diagnosis. Of the 84 patients seeking care for mental symptoms, 35% had a future mental sick leave diagnosis. However, for patients perceiving high work-related stress, no association was found between the reason for consultation and having a future musculoskeletal sick leave diagnosis. In other words, people with high stress did not seek care for musculoskeletal reasons, nor did they receive a musculoskeletal sick leave diagnosis. It cannot be ruled out that the non-significant findings are caused by a lack of statistical power, but the trends indicate no association. To the authors knowledge, the association between the patient reported reason for consultation and diagnosis-specific sick leave has not been researched before. Månsson et al. [52] showed that among Swedish primary health care patient both the reason for consultation and the subsequent diagnosis were most often related to musculoskeletal, respiratory and circulatory ill health. A related Danish study authored by Rosendal et al. [53] using slightly different measures, showed that more than ninety percent of the patients with a psychological GP-stated reason for consultation were classified in the psychological chapter of the International Classification of Primary Care (ICPC) [54], that is a classification adapted for primary health care. However, using patient's reasons for consultation or the GP-stated reasons for consultation can affect the association between the factors under study. A German study by Kratz et al. [55] showed that in patients judging themselves to be depressed, the GPs diagnosed depression in only 39% of cases. One possible reason for these differences is that GPs have a strong biomedical focus during consultation [53], which might have affected the association between the parameters analysed in this study.

Ill health due to work-related stress was in this study viewed as evolving over time. Along this illness-sickness-sick leave trajectory, positions and decisions are made, which can change its course. For instance, studies have shown that, apart from the illness perceived, modifiable factors inform the decision to consult a GP, such as the patient's perceived efficacy of both self-care and GP care, health beliefs, cues to consult, the need for information and impact on daily living [56–58]. Further, although, the patient's and GP's assessment of reduced work capacity are strong predictors for deciding on issuing a sickness certification [59] other factors than work capacity could influence the decision to certify sick leave. Since the GPs find it difficult to assess work capacity and perceive it as not within their purview [60], the decision to certify sick leave may rest on the diagnosis [16]. In addition, GPs might choose the diagnostic code that best corresponds to the patient's health needs when issuing sick leave, that is a diagnosis with a recommended length of sick leave [15] judged as appropriate to treat the patient's illness rather than to focus on the reduction in work capacity [12, 61]. Another aspect to consider is that certificates with symptom diagnoses, compared to certificates with disease specific diagnoses, may lack information to a higher degree, resulting in a comparatively poorer quality of the certificate and thus a poorer basis for the decisions made by the sickness insurance officer [62]. There is thus room for interpretation, which can affect the trajectory and thereby the association between work-related stress, reason for consultation and diagnosis-specific sick leave.

Knowledge shaping and the epistemological foundation for the medical practice are also important for the interpretation of the study findings. Medical records shape the medical knowledge by keeping information that is considered relevant and in turn, the medical and scientific knowledge shape clinical practice [11, 63]. The widely used classification systems ICD [17] and ICPC [54] have a strong anchoring in biomedicine [64], which has led to difficulties in incorporating the social circumstances contributing to distress into clinical practice [65]. In addition, the social circumstances at work might not be properly considered, since primary health care not always has the prerequisites needed [66]. Hence, most psychological problems are classified within a few diagnostic categories and social matters are rarely considered or classified [53]. The association between work-related stress, reason for consultation and diagnosis-specific sick leave, can thereby be seen as reflecting the illness-sickness-sick leave trajectory based on the accumulated medical knowledge, but to some degree also the social and cultural understanding of how to assess, describe, treat and care for patients with stress-related ill health.

## Methodological considerations

In this study, the association between work-related stress and sick leave was viewed from a less researched angle, by using diagnosis-specific sick leave as an outcome measure whilst also including the reason for consultation. Moreover, the longitudinal design of this observational study made it possible to interpret the direction of the association.

When interpreting the findings, it is important to consider the generalizability, since the study population included patients seeking care at seven primary health care centres in the Västra Götaland region in Sweden. To strengthen the generalizability of the findings, both private and public run centres as well as centres located in urban areas, rural areas and towns of different sizes were included. It should also be noted that the study population reflected the inclusion criteria for the RCT, which were set to include non-sick-listed employed patients seeking care for symptoms that could potentially be caused by work-related stress. The distribution of symptoms thereby differed from the primary health care patients in general [52]. Further, the generalizability of the findings to other nations depends on the structure of the

health care system. The results are therefore more easily transferable to countries with a state-regulated universal health care system set to provide basic health services to all their citizens similar to the Swedish system [67].

Data on the reason for consultation and the self-assessed work-related stress was collected at baseline, while the sick leave diagnosis was collected over the 12 subsequent months. During the study period, additional diseases may have occurred, which could explain some of the variation in the study. In addition, the level of stress was only measured once, but this instantaneous value might not be representative for the patient during the period of interest. It is also important to have in mind that this study only includes the diagnosis reported in the sickness certificate, although comorbidity is common for mental disorders [13]. The patient may therefore have other diagnosis or non-diagnosed ill health that could increase the understanding of the relatedness between work-related stress, reason for consultation and diagnosis-specific sick leave.

The individual's perception and experiences of the work environment was of interest for this study and it was therefore found relevant to use self-reports such as the WSQ. As previously mentioned, the face validity and the test-retest reliability of the WSQ have been found satisfying [44, 52]. By using the WSQ, both contextual aspects and personal characteristics were considered, but other psychosocial work factors than those included in the WSQ could be important. It can therefore not be ruled out that a different questionnaire might have captured the most significant factors more accurately.

The use of registry data on prior sick leave and sick leave diagnosis decreased the risk of dropouts and eliminated risk of recall bias. However, the two sick leave measures were limited in time to one year before and one year after baseline respectively. Choosing a longer period of time could have given more information, since it can take time for stress-related symptoms to develop and patients are known to seek care for various complaints for a long period of time for symptoms known to be stress-related [13, 14]. However, if a longer time period was used, it would have been much more uncertain whether the prior sick leave and the future sick leave diagnosis were linked to morbidity at the time of inclusion.

The sample size was not calculated specifically for this study. Lack of statistical power can therefore not be excluded as a potential explanation for the statistically non-significant associations found in this study, especially when performing stratified analysis. In addition, Lidwall [68] questioned the use of broad diagnosis categories in research. The critique was based on study findings showing that there were distinct differences in return to work within ICD-10 chapters. It can therefore be assumed that a larger sample and a more differentiated categorization of the ICD diagnoses could have increased the understanding of the association between work-related stress and diagnosis-specific sick leave further.

## Conclusions

Compared to those who did not report high work-related stress, patients reporting high work-related stress more often sought care for mental symptoms, sleep disturbance and fatigue, and they had a higher risk of future sick leave with a mental diagnosis. Reporting high work-related stress was, however, not linked to having sought care for musculoskeletal symptoms nor future sick leave with a musculoskeletal diagnosis. Thus, both patients and GPs seem to characterize work-related stress as a mental disorder. In research, using diagnosis-specific sick leave rather than all-cause sick leave and including the reasons for consultation can increase the understanding of the relationship between work-related stress and sick leave. In addition, larger studies with more participants would enable a more detailed categorization of both the diagnoses and the reasons for seeking care.

## Supporting information

**S1 Appendix. The work stress questionnaire including instructions for evaluation.**
(PDF)

## Acknowledgments

The authors would like to acknowledge the primary health care centres, especially the GPs, and the patients taking part in this study, as well as the co-workers in the research group TIDAS for their support and feedback.

## Author Contributions

**Conceptualization:** Anna-Maria Hultén, Kristina Holmgren, Pernilla Bjerkeli.

**Data curation:** Anna-Maria Hultén, Pernilla Bjerkeli.

**Formal analysis:** Anna-Maria Hultén, Pernilla Bjerkeli.

**Funding acquisition:** Kristina Holmgren.

**Investigation:** Kristina Holmgren.

**Methodology:** Anna-Maria Hultén, Kristina Holmgren, Pernilla Bjerkeli.

**Project administration:** Anna-Maria Hultén.

**Resources:** Kristina Holmgren.

**Supervision:** Kristina Holmgren, Pernilla Bjerkeli.

**Validation:** Anna-Maria Hultén, Kristina Holmgren.

**Visualization:** Anna-Maria Hultén.

**Writing – original draft:** Anna-Maria Hultén.

**Writing – review & editing:** Anna-Maria Hultén, Kristina Holmgren, Pernilla Bjerkeli.

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
