## [Decision Letter · Decision Letter 0]

25 Nov 2022

PONE-D-22-26733Work-related stress, reason for consultation and diagnosis-specific sick leave: How do they add up?PLOS ONE

Dear Dr. Hultén,

Thank you for submitting your manuscript to PLOS ONE. After careful consideration, we feel that it has merit but does not fully meet PLOS ONE’s publication criteria as it currently stands. Therefore, we invite you to submit a revised version of the manuscript that addresses the points raised during the review process.

We look forward to receiving your revised manuscript.

Kind regards,

Ramune Jacobsen

Academic Editor

PLOS ONE

Journal Requirements:

Reviewers' comments:

Reviewer's Responses to Questions

**Comments to the Author**

1. Is the manuscript technically sound, and do the data support the conclusions?

Reviewer #1: Yes

Reviewer #2: Yes

2. Has the statistical analysis been performed appropriately and rigorously? 

Reviewer #1: Yes

Reviewer #2: Yes

3. Have the authors made all data underlying the findings in their manuscript fully available?

Reviewer #1: Yes

Reviewer #2: Yes

4. Is the manuscript presented in an intelligible fashion and written in standard English?

Reviewer #1: Yes

Reviewer #2: Yes

5. Review Comments to the Author

Reviewer #1: I would like to thank all authors for their interesting and useful work. The paper was presented in a scientific manner. Introduction ,methodology , results and conclusions are clearly written.

study limitations were identified. If possible can add a flow chart. However authors have already explained the patient recruitment procedure as well.

Reviewer #2: Over all the paper content is great.

1. The text alignment should be in Justify (Ctr+J)

2. Tables format/bordering should be similar and formal.

3. Table citation within the paragraphs should be revised.

4. Line spacing including the headings should be revised well as PLOS requirements.

5. Some figure captions are bold and some are not, please update it.

6. Exaggerated paragraph indentation should be amended.

7. Abbreviations should be written in full statement (General practitioners first term under the abstract written as GPS) hence, please update all this abbreviations issue.

8. Was that not possible to analyze logistic regression for the associations besides the P value?

6. PLOS authors have the option to publish the peer review history of their article (what does this mean?). If published, this will include your full peer review and any attached files.

Reviewer #1: No

Reviewer #2: No

---

## [Author Response · Author response to Decision Letter 0]

6 Mar 2023

Dear Editor,

Thank you for the comments. Based on our understanding of the research field, the problem addressed in the study, and PLOS ONE style requirements, we have adjusted the manuscript accordingly. Changes have been highlighted in the manuscript. In addition, point-by-point responses to the comments are presented below.

Comments from the academic editor

Comment 1: Please ensure that your manuscript meets PLOS ONE's style requirements, including those for file naming. 

Response: Thank you for kindly reminding us of the style requirements. The manuscript has been revised to align with the PLOS ONE style templates and the submission guidelines.

Comment 2. We note that you have indicated that data from this study are available upon request. PLOS only allows data to be available upon request if there are legal or ethical restrictions on sharing data publicly. If there are ethical or legal restrictions on sharing a de-identified data set, please explain them in detail (e.g., data contain potentially sensitive information, data are owned by a third-party organization, etc.) and who has imposed them (e.g., an ethics committee). Please also provide contact information for a data access committee, ethics committee, or other institutional body to which data requests may be sent.

Response: To protect privacy and confidentiality, the raw data and the datasets generated in this study are not publicly available due to restrictions stated in the ethical approval (reference number 125-5) issued by the Regional Ethical Review Board in Gothenburg, Sweden (later merged into the Swedish Ethical Review Authority, https://etikprovningsmyndigheten.se/). According to the approval, the data are to be managed carefully, and the results presented so that individuals cannot be identified. In addition, “All results from the project must be reported at group level with de-identified material” (ethical approval 125-5, section 6.3). The raw data and datasets generated cannot be made publicly available, because of the risk that individuals are identified if study data are combined. Sensitive data on work-related stress, days of sick leave and sick leave diagnosis could then be traced to the persons identified. Interested researchers may send data access requests to the corresponding author (AMH), anna-maria.hulten@gu.se, or the TIDAS research group at the Institute of neuroscience and physiology, tidas@neuro.gu.se. Please refer to the TIDAS project “Early identification of work-related stress”, study "How do they add up?", and data set AddUp2023. 

Comments from reviewer 1

Comment 1: I would like to thank all authors for their interesting and useful work. The paper was presented in a scientific manner. Introduction, methodology, results and conclusions are clearly written. Study limitations were identified. If possible can add a flow chart. However, authors have already explained the patient recruitment procedure as well.

Response: Thank you for the positive comments.

Comments from reviewer 2

Comment 1: The text alignment should be in Justify (Ctr+J)

Response: We have studied the style requirements and looked at several articles published in PLOS ONE, but we cannot find any requirement of adjusting the text to both left and right (justify). We therefore chose not to use this typographic alignment.

Comment 2: Tables format/bordering should be similar and formal.

Response: The table bordering has been adjusted.

Comment 3: Table citation within the paragraphs should be revised.

Response: In the paragraphs, tables are cited as “Table 1”, “Table 2” etc. We have studied the style requirements and looked at a number of articles published in PLOS ONE, but we cannot find any ground for revising the table citations. 

Comment 4: Line spacing including the headings should be revised well as PLOS requirements.

Response: Both body text and headings have been revised.

Comment 5: Some figure captions are bold and some are not, please update it.

Response: The figure captions have been revised in accordance with the style requirements.

Comment 6: Exaggerated paragraph indentation should be amended.

Response: The indentation has been adjusted.

Comment 7: Abbreviations should be written in full statement (General practitioners first term under the abstract written as GPs) hence, please update all this abbreviations issue.

Response: Abbreviations have been adjusted, so that they are defined upon first appearance in the text.

Comment 8: Was that not possible to analyse logistic regression for the associations besides the P value? 

Response: Thank you for this question. In this study, the Pearson chi-squared test and calculation of crude relative risks (RR) were considered suitable statistical methods in data analysis, based on the four research questions on the bivariate relationships between work-related stress, reason for consultation, and diagnosis-specific sick leave. Developing multivariable clinical prediction models for binary outcomes was not the scope of this study. However, multivariate analyses could have been used to analyse the three variables simultaneously. Given that non-parametric tests had to be used, binomial logistic regression could be suitable for analysing the relationship between stress, reason for consultation, and diagnosis-specific sick leave. Due to the relatively low sample size, the power was, however, considered insufficient for performing more complex analyses [1]. The analyses would include a large number of statistical tests, which would influence the sample size needed; the more tests, the larger the sample.

In addition to the above comments, minor grammatical changes and clarifications have been made to improve readability (highlighted in the file labelled 'Revised Manuscript with Track Changes').

We look forward to hearing from you regarding our submission and to respond to any further questions or comments. 

Yours sincerely,

Anna-Maria Hultén, MSc, PhD student

Reference

1. Bujang MA, Sa'at N, Sidik T, Joo LC. Sample Size Guidelines for Logistic Regression from Observational Studies with Large Population: Emphasis on the Accuracy Between Statistics and Parameters Based on Real Life Clinical Data. Malays J Med Sci. 2018;25(4):122-30.

---

## [Decision Letter · Decision Letter 1]

18 Apr 2023

PONE-D-22-26733R1Work-related stress, reason for consultation and diagnosis-specific sick leave: How do they add up?PLOS ONE

Dear Dr. Hultén,

Thank you for submitting your manuscript to PLOS ONE. After careful consideration, we feel that it has merit but does not fully meet PLOS ONE’s publication criteria as it currently stands. Therefore, we invite you to submit a revised version of the manuscript that addresses the points raised during the review process.

We look forward to receiving your revised manuscript.

Kind regards,

Ramune Jacobsen

Academic Editor

PLOS ONE

Reviewers' comments:

**Comments to the Author**

1. If the authors have adequately addressed your comments raised in a previous round of review and you feel that this manuscript is now acceptable for publication, you may indicate that here to bypass the “Comments to the Author” section, enter your conflict of interest statement in the “Confidential to Editor” section, and submit your "Accept" recommendation.

Reviewer #1: All comments have been addressed

Reviewer #3: (No Response)

2. Is the manuscript technically sound, and do the data support the conclusions?

Reviewer #1: Yes

Reviewer #3: Partly

3. Has the statistical analysis been performed appropriately and rigorously? 

Reviewer #1: Yes

Reviewer #3: No

4. Have the authors made all data underlying the findings in their manuscript fully available?

Reviewer #1: Yes

Reviewer #3: Yes

5. Is the manuscript presented in an intelligible fashion and written in standard English?

Reviewer #1: Yes

Reviewer #3: Yes

6. Review Comments to the Author

Reviewer #1: All the comments raised by the reviewer has been addressed by the authors. The authors have rearrange the manuscript in more informative and scientific pattern.

Reviewer #3: 

I think authors did not address comment from reviewer 1. As you see, reviewer 1 asked for a flow chart. So far I understand a flow chart of the study population.

I have some comments on methods:

- Is the study cross-sectional or longitudinal? Although authors mention a cross-sectional and prospective longitudinal study design, I think this needs an explanation how a study can be both? Is it they wanted to see differences between two time points, but that was not the case in this study?

- How authors calculated crude relative risk - is it manually without a regression model? If so, they also calculated CI and p-value. Without having a model the statistical results for CRR could be spurious.

- How authors made an index for WSQ instrument is not clear, and what is the reliability of the instrument in this study.

- Author mentioned about power of the study in answering the comment of Reviewer 2. My question is to author whether they had a power calculation; moreover, I notice they have 232 subjects, in that case they are able to perform some kind of models.

- Minor thing: Authors mention about RCT in method - I understand they were explaining how the data were collected, but it seems they talk so much about RCT. Readers may get confused.

7. PLOS authors have the option to publish the peer review history of their article (what does this mean?). If published, this will include your full peer review and any attached files.

Reviewer #1: No

Reviewer #3: No

---

## [Author Response · Author response to Decision Letter 1]

13 May 2023

Response to reviewers

PONE-D-22-26733, PLOS ONE

Work-related stress, reason for consultation and diagnosis-specific sick leave: How do they add up?

Decision received 18 April 2023

Dear Editor,

Thank you for the additional comments, which we found very valuable. Changes have been highlighted in the manuscript. In addition, point-by-point responses to the comments are presented below.

Comments from reviewer 3

Comment 1: I think authors did not address comment from reviewer 1. As you see, reviewer 1 asked for a flow chart. So far I understand a flow chart of the study population.

Response: Reviewer 1’s comment on adding a flowchart read: “If possible can add a flow chart. However, authors have already explained the patient recruitment procedure as well”. Unfortunately, we gave no response to that comment. The reasoning for not including a flowchart was that the inclusion process only included a few simple steps, which we have, as stated by reviewer 1, described in words. In addition, Table 1 gives further information about the inclusion process. A flowchart of enrolment, allocation and follow-up for the RCT can be found elsewhere (1). 

Comment 2: Is the study cross-sectional or longitudinal? Although authors mention a cross-sectional and prospective longitudinal study design, I think this needs an explanation how a study can be both? Is it they wanted to see differences between two time points, but that was not the case in this study?

Response: Thank you for the valuable comment. The study is a prospective longitudinal observational study with initial cross-sectional analyses. Changes have been made to simplify and reduce the risk of misinterpretation (p 2, row 22; p 7, row 146; p 7, row 156; p 21, row 429; p 26, row 542-543). In addition, figure 1 and 4 have been revised.

Comment 3: How authors calculated crude relative risk - is it manually without a regression model? If so, they also calculated CI and p-value. Without having a model, the statistical results for CRR could be spurious.

Response: Thank you for bringing this ambiguity to our attention. We did not perform regression analysis. The relative risk and confidence interval were calculated using IBM SPSS Statistics version 25.0 (Analyze >Descriptive Statistics >Crosstabs). Therefore, we chose to remove the word crude in order not to lead into the idea that the calculated risk constituted an initial step in a multivariable regression analysis (p 13, row 289; p 17, row 357; p 18, row 368).

Comment 4: How authors made an index for WSQ instrument is not clear, and what is the reliability of the instrument in this study.

Response: We agree with Reviewer 3 that the text on how the dichotomized WSQ values were created had to be clarified and have revised the text accordingly (p 9, row 188-206; p 10, row 207-208; p 10, row 211-212; p 10, row 217-229; p 11, row 230-235). 

Thank you for also addressing the reliability of the WSQ. The questionnaire was developed based on both theory and empirical findings about work-related stressors and stress. The results of a qualitative study on women’s views of opportunities for and obstacles to returning to work (2) significantly affected the questionnaire development. As written in the manuscript, the face validity and the test-retest reliability has been tested for women and men separately (3, 4). A pilot group of ten employed working women and a pilot group of seven employed working men confirmed the face validity of the questionnaire. The confirmative responses from both women and men were based on their comments and notes on scale steps and formulation of the questions as well as if the questionnaire corresponded to their understanding of work-related stress after filling out the WSQ. The test-retest reliability of the WSQ has been assessed in two separate test–retest studies with 52 women and 41 men. The reliability was analysed with a non-parametric statistical method for evaluation of paired data (5). The analysis indicated a satisfactory reliability for both women and men.

The WSQ was used in this study, since it includes aspects relating to work, but also to the individual and the time spent out of work. In addition, it is developed to assess the risk of future long-term sick leave in a primary health care setting. At the time of the study there was no other high-quality questionnaire that included all the aspects of interest when looking at stress-related ill health from a transactional perspective. The Cronbach’s alpha is frequently used in research as a measure of scale reliability. However, there are assumption to when it can be used, for instance that the items should be continuous with normal distributions (6, 7). In our study the data is ordinal and therefore we chose not to include Cronbach’s alpha as a measure of scale reliability (internal consistency) in the manuscript.

The following articles describe the development of the WSQ as well as studies in which the questionnaire has been used (all items or in some cases selected items were used):

• Hultén et al, 2022. Work‑related stress and future sick leave in a working population seeking care at primary health care centres: a prospective longitudinal study using the WSQ (8);

• Hultqvist et al, 2021. Does a brief work-stress intervention prevent sick-leave during the following 24 months? A randomized controlled trial in Swedish primary care (9); 

• Hultén et al, 2021. Self-reported sick leave following a brief preventive intervention on work-related stress: a randomised controlled trial in primary health care (1);

• Sandheimer et al, 2020. Effects of a work stress intervention on healthcare use and treatment compared to treatment as usual: A randomised controlled trial in Swedish primary healthcare (10);

• Hultén et al, 2020. Positioning work related stress - GPs' reasoning about using the WSQ combined with feedback at consultation (11);

• Bjerkeli et al, 2020. Does early identification of high work related stress affect pharmacological treatment of primary care patients? - Analysis of Swedish pharmacy dispensing data in a randomised control study (12);

• Frantz et al, 2019. The Work Stress Questionnaire (WSQ) - Reliability and face validity among male workers (4);

• Holmgren et al, 2019. Does early identification of work-related stress, combined with feedback at GP-consultation, prevent sick leave in the following 12 months? (13);

• Holmgren et al, 2016. Early identification in primary health care of people at risk for sick leave due to work-related stress - Study protocol of a randomized controlled trial (RCT) (14);

• Holmgren et al, 2014. Remain in work-What work-related factors are associated with sustainable work attendance: A general population-based study of women and men (15);

• Holmgren et al, 2013. Early identification of work-related stress predicted sickness absence in employed women with musculoskeletal or mental disorders: a prospective, longitudinal study in a primary health care setting (16);

• Holmgren et al, 2013: The combination of work organizational climate and individual work commitment predicts return to work in women but not in men (17);

• Holmgren et al, 2010. The association between poor organizational climate and high work commitments, and sickness absence in a general population of women and men (18);

• Holmgren et al, 2009. The prevalence of work-related stress, and its association with self-perceived health and sick-leave, in a population of employed Swedish women (19);

• Holmgren et al, 2009. Development of a questionnaire assessing work-related stress in women - Identifying individuals who risk being put on sick leave (3);

• Holmgren et al, 2004. Women on sickness absence - Views of possibilities and obstacles for returning to work. A focus group study (2).

Comment 5: Author mentioned about power of the study in answering the comment of Reviewer 2. My question is to author whether they had a power calculation; moreover, I notice they have 232 subjects, in that case they are able to perform some kind of models.

Response: Thank you for raising these questions. A power analysis was performed for the primary outcome measure of the RCT (14). No power analysis was performed for this study, since the study was based on data from the RCT and the number of participants was therefore limited to those patients participating in the trial.

In all 232 patients participated in the study. However, not all patients perceived stress. For instance, 49 patients perceived high stress due to indistinct organisation and conflicts, and 105 patients perceived stress due to individual demands and commitment (Table 5). In addition, not all patients were on sick leave with a mental or musculoskeletal diagnosis (Table 5). Therefore, for some analysis Fisher’s exact test was used instead of Pearson chi-squared test to calculate p-values (the expected cell frequency was less than 5 in more than 20% of the cells in the contingency table). Due to these circumstances, the power would have been to low to perform multivariable analysis. 

Comment 6: Authors mention about RCT in method - I understand they were explaining how the data were collected, but it seems they talk so much about RCT. Readers may get confused.

Response: Thank you for the valuable comment. We have now described the RCT in less detail and rearranged the text as well (p 7, row 149-165; p 8, 166-170). 

In addition to the above comments, minor grammatical changes and clarifications have been made to improve readability (highlighted in the file labelled 'Revised Manuscript with Track Changes').

We look forward to hearing from you regarding our submission and to respond to any further questions or comments. 

Yours sincerely,

Anna-Maria Hultén, MSc, PhD student

References

1. Hultén AM, Bjerkeli P, Holmgren K. Self-reported sick leave following a brief preventive intervention on work-related stress: A randomised controlled trial in primary health care. BMJ Open. 2021;11(3).

2. Holmgren K, Dahlin Ivanoff S. Women on sickness absence--views of possibilities and obstacles for returning to work. A focus group study. Disabil Rehabil. 2004;26(4):213-22.

3. Holmgren K, Hensing G, Dahlin-Ivanoff S. Development of a questionnaire assessing work-related stress in women - Identifying individuals who risk being put on sick leave. Disabil Rehabil. 2009;31(4):284-92.

4. Frantz A, Holmgren K. The Work Stress Questionnaire (WSQ) - Reliability and face validity among male workers. BMC Public Health. 2019;19(1).

5. Svensson E. Ordinal invariant measures for individual and group changes in ordered categorical data. Stat Med. 1998;17(24):2923-36.

6. McNeish D. Thanks coefficient alpha, we'll take it from here. Psychol Methods. 2018;23(3):412-33.

7. Gadermann AM, Guhn M, Zumbo BD. Estimating Ordinal Reliability for Likert-Type and Ordinal Item Response Data: A Conceptual, Empirical, and Practical Guide. Pract Assess Res Eval. 2012;17:1-13.

8. Hultén AM, Bjerkeli P, Holmgren K. Work-related stress and future sick leave in a working population seeking care at primary health care centres: a prospective longitudinal study using the WSQ. BMC Public Health. 2022;22(1).

9. Hultqvist J, Bjerkeli P, Hensing G, Holmgren K. Does a brief work-stress intervention prevent sick-leave during the following 24 months? A randomized controlled trial in Swedish primary care. Work. 2021;70(4):1141-50.

10. Sandheimer C, Hedenrud T, Hensing G, Holmgren K. Effects of a work stress intervention on healthcare use and treatment compared to treatment as usual: a randomised controlled trial in Swedish primary healthcare. BMC Fam Pract. 2020;21(1):133.

11. Hultén AM, Dahlin-Ivanoff S, Holmgren K. Positioning work related stress - GPs' reasoning about using the WSQ combined with feedback at consultation. BMC Fam Pract. 2020;21(1):187.

12. Bjerkeli PJ, Skoglund I, Holmgren K. Does early identification of high work related stress affect pharmacological treatment of primary care patients? - analysis of Swedish pharmacy dispensing data in a randomised control study. BMC Fam Pract. 2020;21(1):70.

13. Holmgren K, Hensing G, Bültmann U, Hadzibajramovic E, Larsson MEH. Does early identification of work-related stress, combined with feedback at GP-consultation, prevent sick leave in the following 12 months?: a randomized controlled trial in primary health care. BMC Public Health. 2019;19(1):1471-2458.

14. Holmgren K, Sandheimer C, Mardby AC, Larsson ME, Bultmann U, Hange D, et al. Early identification in primary health care of people at risk for sick leave due to work-related stress - study protocol of a randomized controlled trial (RCT). BMC Public Health. 2016;16(1):1193.

15. Holmgren K, Löve J, Mårdby AC, Hensing G. Remain in work--what work-related factors are associated with sustainable work attendance: a general population-based study of women and men. J Occup Environ Med. 2014;56(3):235-42.

16. Holmgren K, Fjällström-Lundgren M, Hensing G. Early identification of work-related stress predicted sickness absence in employed women with musculoskeletal or mental disorders: a prospective, longitudinal study in a primary health care setting. Disabil Rehabil. 2013;35(5):418-26.

17. Holmgren K, Ekbladh E, Hensing G, Dellve L. The combination of work organizational climate and individual work commitment predicts return to work in women but not in men. J Occup Environ Med. 2013;55(2):121-7.

18. Holmgren K, Hensing G, Dellve L. The association between poor organizational climate and high work commitments, and sickness absence in a general population of women and men. J Occup Environ Med. 2010;52(12):1179-85.

19. Holmgren K, Dahlin-Ivanoff S, Björkelund C, Hensing G. The prevalence of work-related stress, and its association with self-perceived health and sick-leave, in a population of employed Swedish women. BMC Public Health. 2009;9;73.

---

## [Decision Letter · Decision Letter 2]

5 Jul 2023

Work-related stress, reason for consultation and diagnosis-specific sick leave: How do they add up?

PONE-D-22-26733R2

Dear Dr. Hultén,

We’re pleased to inform you that your manuscript has been judged scientifically suitable for publication and will be formally accepted for publication once it meets all outstanding technical requirements.

Kind regards,

Ramune Jacobsen

Academic Editor

PLOS ONE

Reviewers' comments:

Reviewer's Responses to Questions

**Comments to the Author**

1. If the authors have adequately addressed your comments raised in a previous round of review and you feel that this manuscript is now acceptable for publication, you may indicate that here to bypass the “Comments to the Author” section, enter your conflict of interest statement in the “Confidential to Editor” section, and submit your "Accept" recommendation.

Reviewer #3: All comments have been addressed

2. Is the manuscript technically sound, and do the data support the conclusions?

Reviewer #3: Yes

3. Has the statistical analysis been performed appropriately and rigorously? 

Reviewer #3: Yes

4. Have the authors made all data underlying the findings in their manuscript fully available?

Reviewer #3: Yes

5. Is the manuscript presented in an intelligible fashion and written in standard English?

Reviewer #3: Yes

6. Review Comments to the Author

Reviewer #3: (No Response)

7. PLOS authors have the option to publish the peer review history of their article (what does this mean?). If published, this will include your full peer review and any attached files.

Reviewer #3: No

---

## [Editor Report · Acceptance letter]

10 Jul 2023

PONE-D-22-26733R2 

Work-related stress, reason for consultation and diagnosis-specific sick leave: How do they add up? 

Dear Dr. Hultén:

I'm pleased to inform you that your manuscript has been deemed suitable for publication in PLOS ONE. Congratulations! Your manuscript is now with our production department. 

Kind regards, 

on behalf of

Dr. Ramune Jacobsen 

Academic Editor

PLOS ONE